# COLAEVA: Visual Analytics and Data Mining Web-Based Tool for Virtual Coaching of Older Adult Populations

**DOI:** 10.3390/s21237991

**Published:** 2021-11-30

**Authors:** Jon Kerexeta Sarriegi, Andoni Beristain Iraola, Roberto Álvarez Sánchez, Manuel Graña, Kristin May Rebescher, Gorka Epelde, Louise Hopper, Joanne Carroll, Patrizia Gabriella Ianes, Barbara Gasperini, Francesco Pilla, Walter Mattei, Francesco Tessarolo, Despoina Petsani, Panagiotis D. Bamidis, Evdokimos I. Konstantinidis

**Affiliations:** 1Vicomtech Foundation, Basque Research and Technology Alliance (BRTA), 20009 San Sebastián, Spain; ralvarez@vicomtech.org (R.Á.S.); kmrebescher@vicomtech.org (K.M.R.); gepelde@vicomtech.org (G.E.); 2Biodonostia Health Research Institute, 20014 San Sebastián, Spain; 3Computational Intelligence Group, Computer Science Faculty, University of the Basque Country, UPV/EHU, 20018 San Sebastián, Spain; ccpgrrom@gmail.com; 4School of Psychology, Dublin City University, Glasnevin, D09 X984 Dublin, Ireland; louise.hopper@dcu.ie (L.H.); joanne.carroll@dcu.ie (J.C.); 5Unità Operativa Riabilitazione Ospedaliera—Villa Rosa, Azienda Provinciale per i Servizi Sanitari di Trento, 38123 Trento, Italy; patriziagabriella.ianes@apss.tn.it (P.G.I.); barbara.gasperini@apss.tn.it (B.G.); francesco.pilla@apss.tn.it (F.P.); 6Servizio Ingegneria Clinica, Azienda Provinciale per i Servizi Sanitari di Trento, 38123 Trento, Italy; walter.mattei@apss.tn.it; 7Department of Industrial Engineering, University of Trento, 38123 Trento, Italy; francesco.tessarolo@unitn.it; 8Medical Physics and Digital Innovation Lab, School of Medicine, Aristotle University of Thessaloniki, 541 24 Thessaloniki, Greece; despoinapets@gmail.com (D.P.); pdbamidis@gmail.com (P.D.B.); evdokimosk@gmail.com (E.I.K.); 9WITA SRL, 38123 Trento, Italy

**Keywords:** semi-supervised clustering, visual analytics, coaching, interactive visualization

## Abstract

The global population is aging in an unprecedented manner and the challenges for improving the lives of older adults are currently both a strong priority in the political and healthcare arena. In this sense, preventive measures and telemedicine have the potential to play an important role in improving the number of healthy years older adults may experience and virtual coaching is a promising research area to support this process. This paper presents COLAEVA, an interactive web application for older adult population clustering and evolution analysis. Its objective is to support caregivers in the design, validation and refinement of coaching plans adapted to specific population groups. COLAEVA enables coaching caregivers to interactively group similar older adults based on preliminary assessment data, using AI features, and to evaluate the influence of coaching plans once the final assessment is carried out for a baseline comparison. To evaluate COLAEVA, a usability test was carried out with 9 test participants obtaining an average SUS score of 71.1. Moreover, COLAEVA is available online to use and explore.

## 1. Introduction

Global life expectancy has increased by 5.5 years up to a total of 72 years between 2000 and 2016, and healthy life expectancy has increased by 4.8 years up to 63.3 years according to the World Health Organization (WHO) [1]. Despite living longer, natural age-related physiological decline combined with frequent comorbidities may still require that older adults have some degree of support and monitoring. Furthermore, most older adults prefer living in their own homes instead of residing in residential care, which does not guarantee better outcomes than stay-at-home care [2]. This last point has fostered policies that pursue active and healthy aging, attempting to delay or completely avoid dependency, and allow older adults to enjoy this stage of life as a healthy and independent individual.

Coaching is a form of counseling used to maximize personal potential, according to the ECVision, funded by the European Commission [3]. The coaching strategy states the plan of actions or guidelines. Optimal guidance is achievable when the coach is aware and adaptable to the participant’s context, goals, and preferences. Coaching interventions for older people to maintain healthier habits could prolong their independence and reduce costs to the healthcare systems [3].

In this context, the H2020 Coach Assistant via Projected and Tangible Interface (CAPTAIN) Project is a research and innovation action that attempts to create a virtual coach to support older adults living independently at home. Refs. [4,5,6] CAPTAIN aims to help participants maintain their health and independence by delivering personalized coaching plans that will guide them towards the successful achievement of these goals, as part of their lifelong objectives. Further information regarding this point has been provided in Appendix A.

This paper presents COLAEVA: CAPTAIN—older adult evolution analysis, developed to assist caregivers, in the context of the CAPTAIN project, to manage virtual coaching of a population of older adults, while providing personalized assistance, to:(a)assess the effectiveness of personalized coaching plans(b)identify missing aspects that require coaching in the population(c)design specific coaching plans following a fair-comparison approach(d)suggest coaching plans to new participants based on past experiences collected from prior participants.

This study is based on the topics smart living, behavior understanding, pattern analysis, machine learning, artificial intelligence, unsupervised classification, evolution analysis and human-computer interfaces.

This paper describes COLAEVA, which is available online for use and experimentation [7]. COLAEVA includes several linked interactive visualizations for non-technical users as it is designed for formal caregivers, researchers and primary end-users (to refer to COLAEVA end-users, we will use the term “user” throughout this study). We show COLAEVA’s usefulness and workflow through a use case, presenting several preliminary insights obtained after using the tool on a real and anonymized dataset collected during the first half of the CAPTAIN Project. These preliminary results are not generalizable, because of the data size. Therefore, in order to see how well COLAEVA addresses the needs of these users in a useful way, we have conducted a usability test of the platform. To our knowledge, this is the first time a tool of this kind is presented in the literature. However, there are several recent studies in which clustering in older adults has been found to be effective in studying different targets [8,9,10,11], proving clustering’s potential as part of the COLAEVA feature. These studies, aside from not being in the same field, are carried out on a specific database, and therefore they lack the characteristic of finding new knowledge or searching for it with their own criteria, which is an aspect that we are able to add through COLAEVA.

The remainder of the paper is organized as follows. In Section 2—COLAEVA Platform we have provided a detailed explanation of the platform: the data processing of the various databases used to guide the design as well as the graphical visualizations implemented from two different points of view: (1) individually and (2) grouped into the final application. Then, Section 3—Use Case presents an example of COLAEVA application. Section 4—Usability Test analyses the usability of COLAEVA in order to determine how it is received and in Section 5—Discussion we discuss the results obtained. In Section 6—Conclusions we summarize our conclusions. Finally, in Section 7—Future Work we list the next steps.

## 2. COLAEVA Platform

This section describes the COLAEVA platform, and it is structured following a bottom-up approach, first introducing data processing, then the interactive charts and finally, the visual analytics tool itself.

### 2.1. Data Processing

#### 2.1.1. Data Pre-Processing

The dataset is the result of an observational pilot study conducted over five months with an older adult cohort (*n* = 68) from 4 European pilot sites (Appendix A provides details of the study protocol). To refer to these older adults, we will use the term “participants” throughout this study. The dataset contains information to evaluate the participants’ activities and capabilities in the physical, nutritional, and cognitive coaching dimensions. Social activities were also collected, but no social assessment was carried out, so the relationship between social activities and assessment cannot currently be evaluated, though it is possible to analyze whether social activities have led to an improvement in any other coaching dimension.

This study contains two bodies of data. The first one includes those collected from the caregiver-supervised assessment carried out at the beginning of the pilot period, and those collected five months later, at the end of the pilot period, for all the coaching dimensions (i.e., physical, cognitive, and nutritional). The purpose of the study and data collected is to measure the evolution between the beginning and the end of the pilot period. It contains the scores for several standard questionnaires for each domain:Physical: Fullerton Functional Fitness assessment [12,13,14]Cognitive: Montreal cognitive Assessment (MoCA) [15], Memory complaint scale version A (MCS-A) [16], self-reported and Metacognitive Awareness Inventory (MAI) [17]Nutritional: Mini Nutritional Assessment (MNA mini) Nestle Nutrition, Vevey, Switzerland [18].

The second body of data comes from the daily informal self-reported questionnaires which aim to evaluate the amount and intensity of activities carried out by the participants, regarding the three coaching dimensions.

This pilot study was carried out as part of the research work performed during the H2020 project CAPTAIN, Grant Agreement Number 769830. All details regarding the cohort and data collection are described in Appendix A.

#### 2.1.2. Assessment Data

To estimate the condition of the participants at the beginning and the end of the program, participants underwent several tests whose scores can be directly related to three of the four main coaching dimensions of the program (Nutritional, Physical, Cognitive) each of which are presented in Table 1.

#### 2.1.3. Daily Data

Table 2 presents the specific questions that each participant answered every day. These questions are associated with the four coaching dimensions we followed in the study (Nutritional, Physical, Cognitive and Social). The weights assigned in the table are the level of importance given to each variable within its coaching dimension (if it is negative, the impact it has is negative as well).

The variables in the evolution dataset have been aggregated based on the coaching dimension they belong to, in order to easily compare this dataset with the assessment dataset which is also aggregated at the coaching dimension level. This grouping has been conducted according to the weights assigned in Table 2 (selected empirically, according to the know-how of the experts in the field) which represent the importance we give to each variable within its dimension (if it is negative, the impact it has is negative). The qualitative values were translated into quantitative ones (e.g., answers Yes/No were coded as 1/0). In the questions where there are more than two possible ordered values for the answers, the smallest value has been assigned the value 0, and the others have been assigned the appropriate value considering their ranking order. With this information, the daily activity level of a dimension is estimated as follows:(1)$$dimension\_activity\_level_{day}=\overset{n}{\underset{i=1}{\sum }}w_{i}*a_{i}$$
where $w_{i}$ is the weight associated to the question (see Table 2), and $a_{i}$ is the answer index. The i value ranks the questions of the coaching dimension.

#### 2.1.4. Participants Clustering

In this section we present how participants were clustered in COLAEVA. First, we detail the clustering method we have applied to form participant groups and second, how we estimate the similarity between participants belonging to different groups.

#### Grouping

Based on the results of the initial assessment, participants are grouped so that different groups can receive different recommendations, and members of the same group receive similar recommendations. It is important that participants only receive information relevant to their personal needs and goals.

In order to carry out the appropriate grouping, a hierarchical agglomerative clustering method has been used [19]. That is, as a first step, we grouped the two most similar participants, treating them as singleton clusters. Then the next two closest participants are successively joined until the entire dataset is merged into one single group. To calculate how close two groups are, we used the “average” method. This method estimates the average distance between all possible pairs of elements each belonging to a different cluster. By using this method, we can compare unipersonal distances with intergroup distances. How to calculate the distance between two participants is explained in more detail in Section Distance. This distance represents how similar two participants are, i.e., if the distance is 0, the participants are identical, and the larger the distance, the more different they are.

As a result of this iteration, we generated a dendrogram. The higher two participants/clusters meet on the graph, the larger the difference between them, and therefore the more dissimilar they are. In selecting a dissimilarity threshold of our choice, we can cut this dendrogram creating any number of clusters in which the various participants can be grouped.

#### Distance

While carrying out clustering, participants are grouped according to a distance that represents how similar they are. To calculate this distance, we have used the “weighted Manhattan” [20] method for its proven capacity and good results reported in the literature of the state of the art. With this distance the user can assign different weights to the different variables according to the importance given to each variable. For example, if the user wants to group the participants in terms of physical ability, given that the ability to move relatively freely has a significant impact of the ability of a participant to retain their independence [21], he/she can increase the weight of variables associated with the physical aspect. A similar focus can be applied to each domain as required.

Before distance estimation, the variables are standardized, so that one variable does not have more weight than the others simply because it is measured in larger units. By standardizing the data, all variables will have a mean of 0 and a standard deviation of 1, making direct comparison possible. Moreover, when estimating the standard deviation and mean used for standardization, outliers (estimated by Tukey method [22]) have been removed so that they do not bias this standardization.

Let’s consider that we have u, v∈U where U is the group of participants, and $u_{i}$ and $v_{i}$ represent the standardized values of the *i*-th variable i={1, 2, …, n} of the participants u and v respectively, where n is the number of variables. On the other hand, there is $w\in W^{n}$ where $w_{i}=\left\{0,\;0.5,\;1,\;1.5,\;2\right\}$ depending on the defined weight (very low, low, medium, high, or very high) of each variable, we calculate the distance between participants u and v as:(2)$$d\left(u,\;v\right)=\overset{n}{\underset{i=1}{\sum }}w_{i}\left|u_{i}-v_{i}\right|$$

COLAEVA provides sliders to specify these weights (see Figure 1) so that the users can dynamically change the weights of all the variables associated with each coaching dimension. Users can also adjust the weights of each of the variables in a more precise way via a drop-down menu.

#### 2.1.5. Improvement Estimation

As described previously, two assessments are included in this dataset, representing the participant status at both the beginning and the end of the data collection period. With these two assessments we can extract the participants’ overall improvement, e.g., by comparing the different results obtained at both timestamps. Moreover, we can determine which of the participants have improved within specific coaching dimensions using only the related variables.

### 2.2. Data Visualization Charts

After processing the data as described in the previous section, it is ready to be used in the visualization charts. In this section each chart in the COLAEVA tool is described in detail.

#### 2.2.1. Population Assessment Score Distribution Histogram

The first visualization in the platform is a general overview of the data (see Section 2.3.1). Each variable is presented using a histogram chart showing two overlapped distributions, blue for the first assessments and orange for the second. This visualization allows the user to understand the distribution of the population scores for each assessment and to easily check if there has been an overall change in a particular variable.

Figure 2 depicts the example of the MAI attribute (see Table 1). In this case in the first assessment there are many participants with an MAI value between 5 and 5.3 whereas in the second assessment the results overall are higher, which could indicate an improvement.

#### 2.2.2. Population Grouping Using Clustering and Dendrogram Visualization

As explained in previous sections, the first task is to group the participants based on their results in the first assessment. Depending on the weights that the user defines for the distances, the grouping will be different, as depicted in the dendrogram graph shown in the COLAEVA interface. Figure 3 shows an example of the dendrogram in which there is a general group (purple), a smaller group (light orange) and 8 other groups that are composed of between 1 and 4 participants. These last groups can be considered as outliers since they are very different from rest and constitute a small fraction of all the dataset, so it is convenient to remove them from the insight extraction process carried out over the larger groups. In the top part of Figure 3, a slider appears to select the dissimilarity threshold at which the user wants to cut the dendrogram. Depending on the selected threshold value, more or less groups will be created. Therefore, depending on what the user wants to study (e.g., an exact number of clusters), how close the user wants to have the clusters from each other (dissimilarity level) or depending on the shape of the dendrogram, a higher or lower value will be chosen.

#### 2.2.3. Population Grouping Distribution Pie Chart

Once the dendrogram has been cut by selecting the level of dissimilarity, the platform will display the associated identified clusters. A user can easily infer the sizes of the clusters (number of participants per cluster) upon properly interpreting the dendrogram. For higher precision, a pie chart graph is provided in which the user can see the exact sizes of the clusters, the general proportion of each cluster and, moreover, can associate a name with each of the clusters (“Group n: sedentary”) to each color for further identification.

Figure 4 provides the resultant pie chart of the dendrogram from Figure 3. This pie chart shows that the predominant cluster is the purple one, which is Group 0 with a proportion of 52.7% of the total number of participants. By placing the mouse over this section, the user could see the exact number of participants that belong to that cluster. The light orange group is the second largest (Group 3), maintaining the same color of the dendrogram for simplicity (Figure 3). The others can be considered as outliers given their small size and greater dissimilarity compared to most participants.

#### 2.2.4. Per Group Assessment Characteristics Radar Chart

Having identified the relevant clusters and their sizes, the next step is to indicate to the user the relevant characteristics of each cluster in terms of the initial evaluation. For this purpose, a radar chart graph (see example in Figure 5) is presented. The user can see the averages of the variables (standardized, so that the axes are consistent) of interest per cluster for the first evaluation. In this visualization users can dynamically select the individual clusters they want to visualize so they can compare the clusters of interest. This graph provides a general overview of the participants’ initial state per cluster.

In addition, this allows the user to visualize the variables associated with the different fields on which the user chooses to focus. For example, Figure 5 depicts a use case where the variables associated with the physical and cognitive are shown, however, the variables associated with the nutritional dimension are not displayed (selected domains are displayed at the top of the screen). The user indicates which coaching dimensions they would like to see considered in the graph using the tool bar.

#### 2.2.5. Individual Assessment Evolution Parallel Coordinate Chart

As discussed in Section 2.1.5, the participants’ change throughout the course of the program is estimated by comparing results from the initial and final assessments. In Figure 6 we can see how this change is distributed in the axis’ values of each coaching dimension through a parallel coordinates graph. The participants are designated with a color that matches their cluster’s color. Thereby, users can visually identify patterns of improvement (positive values) and worsening (negative values) associated with different clusters. The graph is interactive so that users can change the order of the axes for better visualization and selecting different visualization spaces for each axis. In the example presented in Figure 6, the user has selected participants who have improved cognitively between 0 and 5 and nutritionally between 0 and 2.5. Upon restricting these parameters, visualization has been reduced to only those participants who meet this requirement.

#### 2.2.6. Per Group Activities Compared with Assessment Evolution Line Charts

Finally, the most relevant graph we use allows us to extract recommendations for program participants shows the effort made by the selected coaching dimension (Figure 7, point 4) throughout the program (x-axis, 150 days) for the participants belonging to the selected group (Figure 7, point 1). These participants are assigned with a color according to whether they have improved or worsened based on comparative evaluations at the beginning and end of the program in the selected dimension (Figure 7, point 5). The color code is as follows: green—participants have improved; yellow—participants have remained in the same state; and red—participants have worsened. A participant may experience a decline in one or more coaching dimensions while simultaneously experiencing improvement in another. In addition, the degree of their improvement or decline in each dimension is reflected using different shades of the assigned color. For example, a significant improvement would result in a darker green line whereas a slight decline would generate a light red line.

To simplify the graph by removing the noise throughout the dataset collection duration, the smoothing moving averages method is applied so that the graph delineates the trend around each point instead of the raw value. Thereby, when selecting the “mean” in the smoothing method (see Figure 7, point 3), the moving average method is used which shows the average of “*n*” days around the selected day (this value of “*n*” can be selected in Figure 7, point 2). This way the user sees the trend of each participant throughout the dataset collection instead of each unique day which might result in a very noisy graph. Following the same approach, the maximum or minimum values can also be displayed, which are selectable in Figure 7, point 3.

These parameters are for the user to adjust in search of meaningful patterns in the graph (Figure 7) that can be directly translated into helpful recommendations.

### 2.3. Application

The COLAEVA application has been implemented using Python 3.7 [23] programming language and an open-source web based application framework called Streamlit [24] that facilitates the creation of web applications with complex interactive charts.

The application has two running modes: “Learning” and “Pro”. On the one hand, in “Learning” mode all charts include explanatory texts, so that the application is used more like a dynamic report and promotes learning how to use the tool. On the other hand, the “Pro” mode hides all of the learning tips, delivering a more compact and efficient visualization once the user has gained experience using the tool (in the “Learning” mode).

The application workflow is divided into two steps. In the first step, the user can review the assessment data result distribution among the population using histograms, organized by the coaching dimensions.

In the second step, the user can distribute the population in different groups selecting the relevance of each coaching dimension for the grouping, and evaluate the properties of each group:Amount of peopleAssessment performance per coaching dimensionAssessment evolution per individualLink between activities on each coaching dimension and the evolution in the assessment data

#### 2.3.1. Step 1: Review Assessment Data

This is the first page in which COLAEVA and the datasets used in the tool are described. In addition, this section provides an overview of the various assessment datasets.

To portray the characteristics of the data in a simple way, a series of distribution functions and histograms are graphed together to show the evolution of participant performance between the initial and final assessment (see Figure 2). In addition, these graphs enable the comparison of the two assessments. In addition, general patterns (in all participants) of improvement or worsening can be observed after follow-up. These graphs are separated by the coaching dimension to which they are associated.

#### 2.3.2. Step 2: Group Users and Study Evolution

This section is the core of the tool. It is comprised of two main steps: first, it groups participants according to their similarities based on the first assessment, and second, it analyses their activity throughout the program and the impact that this activity had on improving or worsening in the different coaching dimensions (differentiating according to the groups created in the first step). This step is widely illustrated in the next section through a concrete use case.

## 3. Use Case

In this section we will provide a use case in which we employ COLAEVA for the extraction of different insights.

Let us say that we are working with a group of older adults who are displaying signs of cognitive decline. Although living independently, there is some concern that their cognitive impairment may impact their ability to support their nutritional needs (e.g., forgetting meal times or over-eating as previous meals are not easily remembered). We want to group these participants giving more importance to nutritional and cognitive aspects, and less to the physical. In this case we would set the weights of physical to “low”, meanwhile cognitive, and nutritional are set to “high” as shown in Figure 8, point 1. After analyzing the dendrogram (Figure 8, point 3) and the descriptive graphs (Pie chart, point 4 and Radar chart point 5 in Figure 8) for the different possible dissimilarity levels (moving the slider in Figure 8, point 2) we conclude that the best grouping is made with the dissimilarity level of 0.98.

There are two main groups (see Figure 8, point 4), one (Group 0) with 52.7% of the participants, and the other (Group 3) with 27%. The remaining seven are groups of 1–2 participants, so we can consider these participants as outliers, i.e., participants who are very different from the rest. Therefore, we will focus on these two main groups to look for patterns. To study how these two main groups differ, we focus on the radar chart graph (Figure 8, point 6) and select the variables associated with the nutrition and cognitive coaching dimensions (Figure 8, point 5), since these are the variables of interest in this particular study. As can be seen in Figure 8, point 6, cognitively there is no evident difference in the group averages, but there is an evident difference nutritionally. Upon evaluation, we will consider that Group 2 has a good nutritional level whereas the opposite is the case for Group 0.

Once we have defined the main groups and their characteristics, we want to see if there is any evident difference in improvement between the clusters. Upon analyzing the parallel coordinates, we have not found any general pattern, but we have found different cluster behaviors when analyzing the evolution graphs.

On the one hand, in Group 0, where a main characteristic was the low nutritional level, we have seen that when the participant maintains a good/high nutritional level throughout the program, they demonstrate an improvement physically at the time of the second assessment. This is evident in Figure 9, where the nutritional activity of the participants of Group 0 is shown throughout the program, colored according to the physical improvement: there are a few participants in this cluster who have worsened physically, colored in red, who are generally lower in the graph (i.e., they have had generally lower nutritional activity during the program) than the rest who are green colored (those who have improved physically). This suggests that not only does nutritional coaching improve the eating habits of this group, but it also has a positive impact on their activity levels. Taken together, the results for the participants in this group indicate that the changes they have made in response to the coaching supports their ability to maintain their independence.

On the other hand, in Figure 10 we show the same graph but for participants in Group 3, where the initial nutrition level was higher. In this case, we do not see the same improvement in physical activity as was found in Group 0. This suggests that these participants were already maintaining good levels of nutrition which positively support their physical abilities. As a result, concentrating on nutritional coaching for this group is unlikely to have significant benefits. The intervention (goals) for this group should be adjusted so that they focus on areas in which the participants have more to gain.

In the graphs of Figure 9 and Figure 10 we can see the usefulness of grouping patients first, since different recommendations are obtained depending on the group to which they belong. On the other hand, we also have the possibility to analyze the overall pattern as shown in Figure 11. Although social-related assessments were not a feature of study underpinning this use case, it is possible to see a general pattern in which people who have maintained higher social activity have improved physically, especially if the higher social level has been maintained throughout the entire duration of the program. Again, this illustrates the value of being able to cluster participant change over time in each of the domains.

To sum up, in this section we have described how the tool can be used and some insights that can be extracted with the current data. It is expected that the amount of data will grow as more users participate in this program and therefore the insights that will be extracted will be more robust.

## 4. Usability Test

We have conducted a formal usability test with a set of test subjects matching the target tool user profiles as has been done in the literature for similar platforms [25,26]. These individuals had no prior experience with the tool to avoid bias in their evaluation of COLAEVA. They were involved in the follow-up of the participants in the CAPTAIN program, so they had expertise in coaching interventions and systems. We will refer to these test subjects with whom we have conducted the usability test as “subjects” throughout the study, in order to maintain consistency in the terminology and to differentiate them from the older adult participants.

The usability test has been carried out with 9 subjects. This number is considered higher than the recommended by the usability experts to maximize cost/benefit ratio of evaluations [27], as 5 subjects can identify 80% of the problems in a software application [28].

### 4.1. Interview Design

The usability test was conducted with nine subjects (1 male and 8 female). Most subjects were either psychologists (3) or researchers (3). There was also a physician, an assistant professor, and a university lecturer. Most were between the ages of 25–40 (6) some between 40–50 (2) and one between 50–60. In addition to the usability test, as an initial step, we collected demographic information about each subject’s level of exposure to digital technologies to better evaluate the final results according to their expertise. These questions are summarized in Table 3 with their respective answers sorted in decreasing order to prevent possible identifications.

As can be seen in Table 3, Q1, most of the subjects are regular computer users (more than half of them spend more than 50 h per week on a computer, three between 30–40 h per week and finally one with low usage with around 10 h per week, with a mean of 42.8 h). In general, according to the answers given to Q2 all subjects seem to be competent computer users. In addition, answers to Q3 show a common standard knowledge of visualization applications. Based on these results, it is expected that the subjects will face no major difficulties interacting with the tool.

Interviews began with a brief overall description of the CAPTAIN project as well as the purpose of the tool and the interview. After the introductory explanation, they were instructed on how to use the tool and the features were explained in extensive detail along with an example for better understanding. After this process, subjects were invited to take control of the tool with the purpose of understanding the older adult population data and propose their own recommendations for the older adult groups. During the interview, subjects were encouraged to follow the Think-Aloud Protocol (TAP) [29] while they interacted with the application freely to better understand the cognitive processes that contributed to their decisions. They were encouraged to ask questions and communicate any insight they were able to identify while playing with the tool. As mentioned in the usability test finding section and, in the discussion, these last two approaches have allowed us to analyze the usefulness of the tool in greater depth and identify possible improvements.

After having used the application for about 15 min, subjects were invited to conclude the test by filling out a final questionnaire. This comprised of a combination of the System Usability Scale (SUS) [30] and a set of custom questions to collect feedback about the tool’s features and insights obtained from the older adult population data depicted in the application. Section 4.2 will review all the answers given to the SUS questionnaire while the answers to the custom questions will be assessed in Section 4.3.

### 4.2. SUS Questionnaire

The SUS score evaluates the effectiveness, efficiency, and overall ease of use of the platform. The collected answers are shown in Table 4.

For the SUS test shown in Table 4, the tool obtained an average score of 71.1, which corresponds to the “Good” qualification according to the SUS mapping [30]. If we analyze the scores per subject, two reported low values (Q0 47.5 and 62.5) which were considered to be nearly acceptable according to Brooke’s estimation [30]. Nevertheless, the rest of the subjects reported good performance in terms of effectiveness, efficiency, and overall ease of use. What is more, one subject reported excellent performance (Q0 82.5).

If we were to perform a deeper analysis, we could conclude that most of the subjects consider that they would use this application in a real-world scenario (In Table 4 for Q1, the majority are 4 with some answers above and below). And likewise, subjects generally agree that it is an easy-to-use application (Q3, mean 4).

Responses were very good in terms of the integration of different components of the platform (Q5, mean 4.2), and lack of inconsistencies (Q6, mean 1.9). COLAEVA is perceived as very comfortable to use (Q8 all responses between 1–2, mean 1.6). In terms of confidence in its usage overall there is a good response (Q9, mean 3.7).

However, although most subjects feel that it has low complexity, there are two subjects who do consider the tool unnecessarily complex (Q2) and who believe that they would need support in using it (Q4). Most subjects believe that they would not need to learn many concepts before using the platform, although there is one subject who thinks they would need to learn a lot and another one who thinks they would need to learn something (Q10, most with values 1–2, one with a 5 and one with a 3). This may occur due to the mathematical methods employed by the tool. Understanding these methods requires that the user has a certain background and knowledge base within this field even though the results are being shown in the simplest possible manner. However, it is expected that with the recurrent use of the application, subjects will understand these methods, making the application more accessible for them over time. In any case, results show that in general subjects believe that people would quickly learn how to properly use COLAEVA.

### 4.3. Personalised Questionnaire

In this section we present the responses to the questions that specifically evaluate the platform, which were complementary to the previously presented SUS questionnaire. In Table 5 we show these questions and their answers. These questions were designed to assess specific features of COLAEVA.

In Table 5 we can observe that the tool obtained a high score for the first question (Q1, mean 4.4), meaning that subjects obtained a very good understanding of how the grouping of older adults is performed in the platform. Similarly, almost everyone believes that the graphs help to properly differentiate the groups created with the application (Q2). As for the evolution graphs, the results were also very high, with all the answers being 4 or more (Q3). For the last question (Q4), even if there was less agreement among subjects, the average result was good (3.8).

### 4.4. Usability Test Findings

In general, the feedback received in the usability test indicates that the tool has been well received by the subjects and that they find it useful, although there were a couple of subjects who felt less comfortable using the platform. In general, there were very good results regarding the presentation and usability of the application. In contrast, results relating to the understanding of some functionalities were lower. This may be due to some mathematical features that we have introduced, that in principle may be more difficult to understand. In addition, the handling of real-world data sometimes complicates the graphs, causing them to be complex to interpret. This was specifically highlighted thanks to the think-aloud protocol. Subject narratives demonstrated that they found the parallel coordinate was difficult to understand.

It has also been found that it is a platform with a learning curve. Subjects commented that it is not a platform that anyone can just pick up and start using without any explanation, but that it requires an adequate explanation and a bit of time to learn how to use it properly. The “Learning mode” is provided to address each of these issues, so that when the user does not know about some of the COLAEVA functionalities, they can easily access a detailed explanation. In this usability test, due to time limitations, the subjects were not able to spend much time using this feature, but it is expected to improve users’ understanding of COLAEVA. Once all the steps were understood, it has been generally commented that it is very useful in the field for which it has been designed.

The response to the platform-specific questionnaires showed very good results and the subjects appreciated the way the data is exploited, how participants are grouped and how the platform reveals insights about participant clusters. Subjects were able to see how the use of the platform could support the design of tailored coaching interventions for specific categories (clusters) of participants.

## 5. Discussion

In this study we have contributed to the extraction of recommendations for the coaching of older adults to improve their quality of life. To do this, we have developed COLAEVA which makes use of retrospective data, machine learning and visualization techniques for recommendation extraction. Although through the usability test, we have seen that the COLAEVA platform alone has a strong utility, its utility improves within the H2020 CAPTAIN project, the context in which it has been developed. Within the CAPTAIN project, COLAEVA focuses on supporting the caregivers when creating the personalized coaching plan for the program participants and it focuses on specific domains (Nutritional, Physical, Cognitive and Social) that are addressed by the CAPTAIN project. A caregiver or older adult family member can access COLAEVA tool in order to gain insights that can help them (1) choose the most appropriate coaching plan and goal for the older adult, (2) create an entirely new coaching plan adapted to a specific cluster of users, (3) adapt an existing plan to the users’ current needs.

Currently the tool can be used to support caregivers in designing coaching plans for each participant, but ideally, it would generate different participant-profiles (clusters) and, automatically propose recommendations for each different cluster. However, this step requires a larger database to make these recommendations more robust and generalizable. This is a problem already identified in the current study and can be addressed properly in future studies collecting larger datasets as COLAEVA tool can easily support the addition of new datasets. Existing literature [31] analyzes also coaching patterns used for older populations and should be carefully considered for the automatic generation of coaching plans. A coaching plan should reflect both the participants needs, as addressed by their data in COLAEVA, but also include the view of professionals. For that reason, COLAEVA is envisaged as a decision support tool for caregivers that want to create coaching plans for older adults. COLAEVA tool can correspond to the “Goal Setting of Target Wellness-Status” tool proposed in the holistic electronic coach (eCoach) model by Gerdes et al. [32] that aims to take into account professional guidelines, existing knowledge from user and the user’s needs and preferences in the creation of a goal plan.

In addition, there are still some areas where we can improve COLAEVA’s capabilities. For example, in the usability test, a subject realized that as soon as a participant enters and begins to follow the given recommendations, he/she may change clusters (they can improve in an assessment, so they would change cluster and hence, would better fit the updated recommendations). Given this possibility, ideally these participants would be assessed more often and would therefore receive the best possible updated recommendations.

Finally, it was clear from the examples presented, that not all participants can be allocated to clusters. These have been referred to thus far as outliers. These outliers reflect participants who do not conform to the general patterns of behavior across the domains of interest that are often the ‘noise’ in the data and discounted when analyzing all participants in a dataset. In fact, these are often the participants with more complex behavioral interactions across domains that require personalized assessment and interventions in order to achieve successful outcomes. The COLAEVA tool quickly identifies who these participants are. Given that interventions can be easily and quickly tailored for participants in the dominant clusters, this leaves researchers with more time to focus on tailoring interventions for these more ‘different’ individuals. Taking this a step further, for example into clinical practice, means that the COLAEVA tool could provide clinicians with the ability to easily tailor intervention programs to suit the needs of the individual.

## 6. Conclusions

This paper presents COLAEVA: CAPTAIN—older adult evolution analysis, developed for caregivers to manage virtual coaching for older adults, focusing on (a) assessing the effectiveness of personalized coaching plans (b) identifying missing aspects requiring coaching in the population thereby supporting the design of specific coaching plans following a fair-comparing approach and (c) suggesting coaching plans to users based on past experiences of other users.

The combination of the first feature of grouping users with semi-supervised classification methods with the second feature of evolution charts has proved to be useful in the example we have analyzed. In this example we have seen that the participants who a priori have a low nutritional level are likely to improve physically if they maintain a high quality of nutrition (see Section 3). In addition to this combination, we have seen that the evolution graph is able to extract insights on its own without previously grouping the participants (i.e., using all participants). For example, patients who maintain a high socialization level also improve physically, especially if this high social interaction is maintained shortly before the second assessment (Figure 11).

A usability test has been performed to assess the usability and capacities of COLAEVA. With this usability test, we have concluded that the tool has been very well received especially for its simplicity and the way in which it displays the graphs. As a drawback, several users have reported a learning curve which, in our opinion, could be due to the complex methods it makes use of. Consequently, it is not straightforward to properly accomplish all the steps required to extract recommendations without having previously acquired a certain amount of expertise. For this reason, some of the users have felt that they would need assistance to properly use the tool and would need to better understand a few concepts before they could use it correctly. These concerns can be addressed through the provision of detailed use cases with supporting audiovisual illustration of how to use COLAEVA to understand how the use case data can support decision-making. Nevertheless, a good (according to the thresholds defined by Brooke [30]) SUS score has been achieved considering the complex techniques it deals with. As for the platform-specific questions, they have been very positive and test subjects generally found the platform to be useful.

In conclusion, positive feedback and constructive recommendations for the future have been obtained from the platform so that we can improve the platform. A possible improvement would be to increase the number of clusters or to change the radar chart graphic for a simpler one. In general, it is expected that the implementation of these recommendations in combination with a higher number of participants would lead to a tool that could ultimately be integrated into the caregivers’ daily routine. Furthermore, the ease with which it can cluster participants and identify those outliers that require more detailed attention, demonstrates the value that the tool can bring to clinical research and practice.

## 7. Future Work

At this point, we have defined some next steps. First, we have the possibility to improve some technical aspects of the tool itself based on the findings of the usability test. For example, as we have commented in the discussion, the parallel coordinate was found to be difficult to understand by those who are not used to working with this type of visualization, therefore, we will study if other graphs are more suitable to display that information. Secondly, we see the need to increase the number of participants to extract more robust and generalizable recommendations in order to ratify and compare our results with the recommendations of the state of the art [31,33]. Therefore, we will look for new potential participants to increase the database. Moreover, we consider including, in addition to graphical evidence, statistical tests for group comparisons, outlier detection, etc. Finally, we aim to use COLAEVA (in the scope of CAPTAIN) with real people, so that, those being supported to live independently by family, friends and formal caregivers, in order to improve their situation and to support them to live well at home for longer.

Furthermore, in future versions of COLAEVA tool, data from real-life monitoring inside a home from CAPTAIN system, can be added. These data are collected in a way to correspond to the ones fed in COLAEVA tool and can be used for creating new coaching plans or to adapt the existing ones in order to respond to changes in users’ behavior and adapt to the behavior change curve.

## Figures and Tables

**Figure 1 sensors-21-07991-f001:**
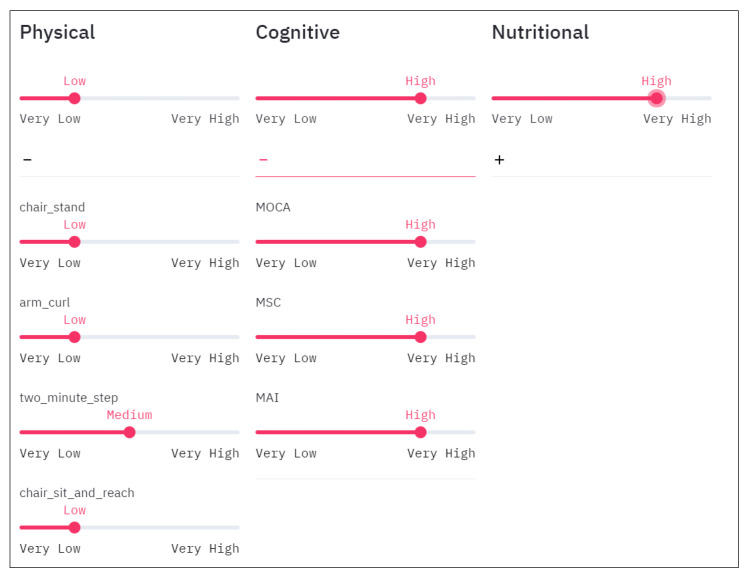
Sliders to specify different weights (importance) to variables for distance calculation.

**Figure 2 sensors-21-07991-f002:**
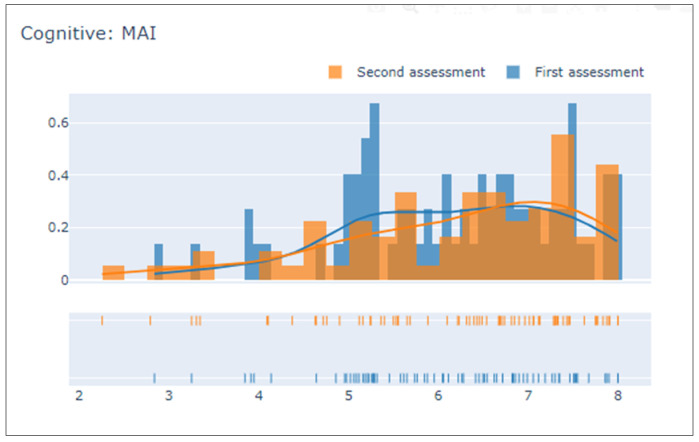
Description of MAI variable performance: first vs. second assessments.

**Figure 3 sensors-21-07991-f003:**
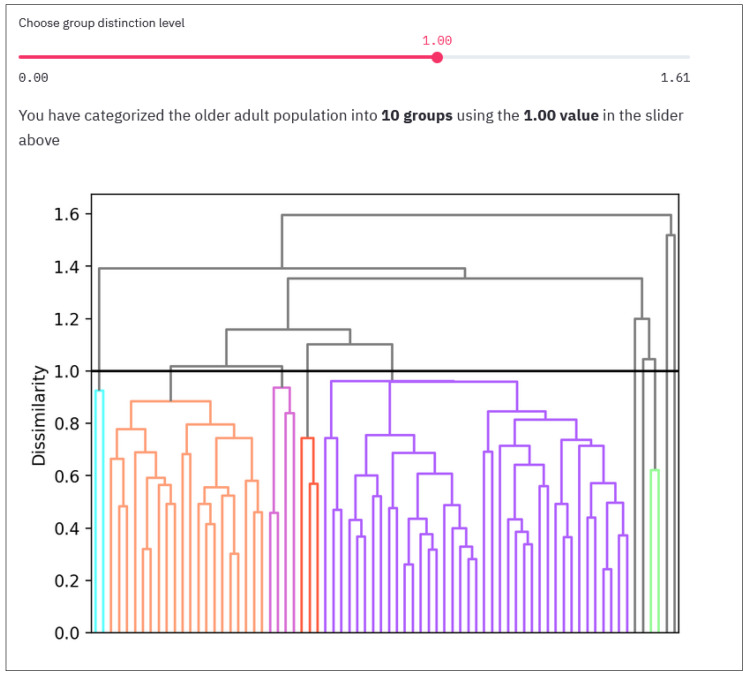
A dendrogram to create the grouping with its slider to select the dissimilarity level.

**Figure 4 sensors-21-07991-f004:**
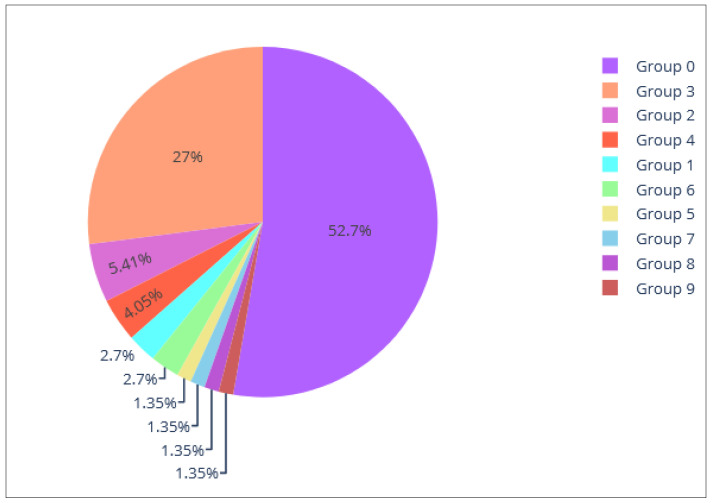
Pie chart of the example shown in Figure 3.

**Figure 5 sensors-21-07991-f005:**
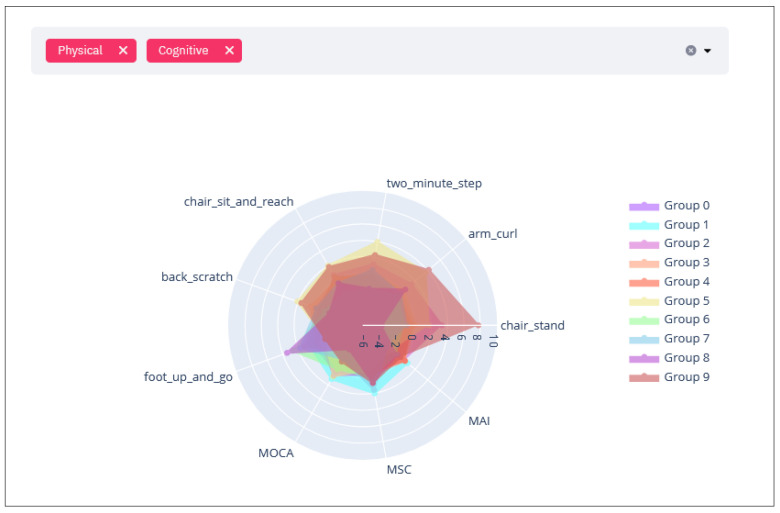
Radar chart of the clusters obtained in Figure 3. In this case it shows the variables associated with the Physical and Cognitive fields.

**Figure 6 sensors-21-07991-f006:**
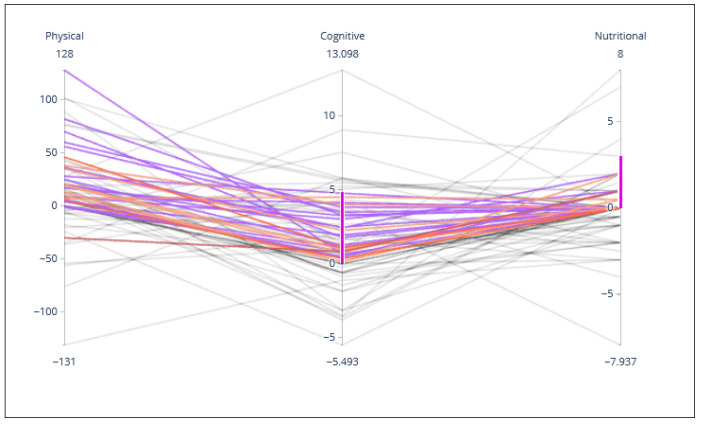
Changes in each field per participant shown in a parallel coordinate, determined by the color of the cluster the participant belongs to.

**Figure 7 sensors-21-07991-f007:**
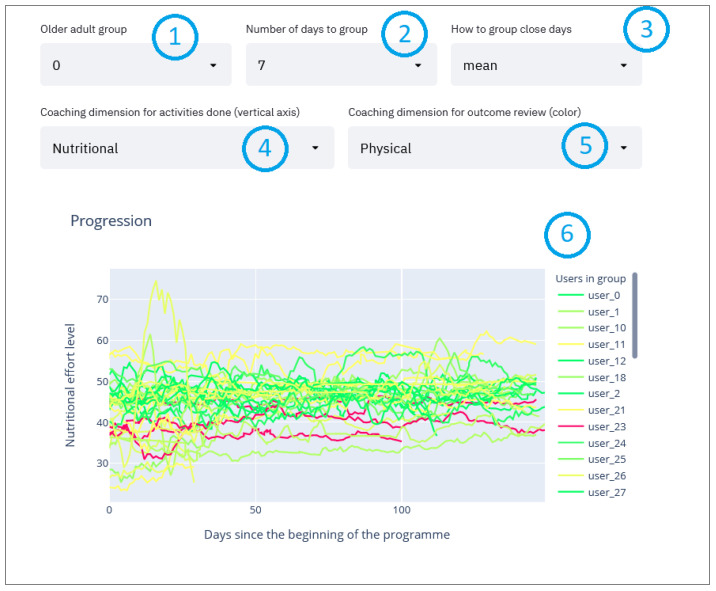
Evolution graph.

**Figure 8 sensors-21-07991-f008:**
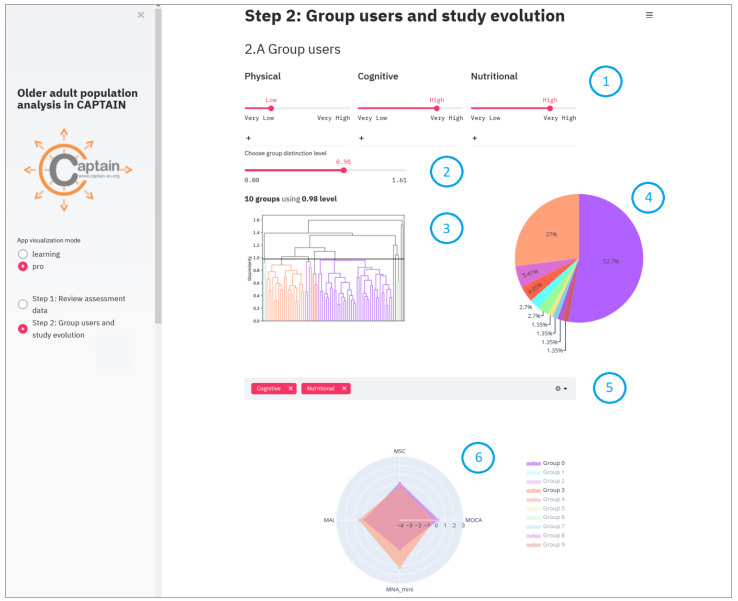
A use-case in Pro mode and Step 2, part Group users.

**Figure 9 sensors-21-07991-f009:**
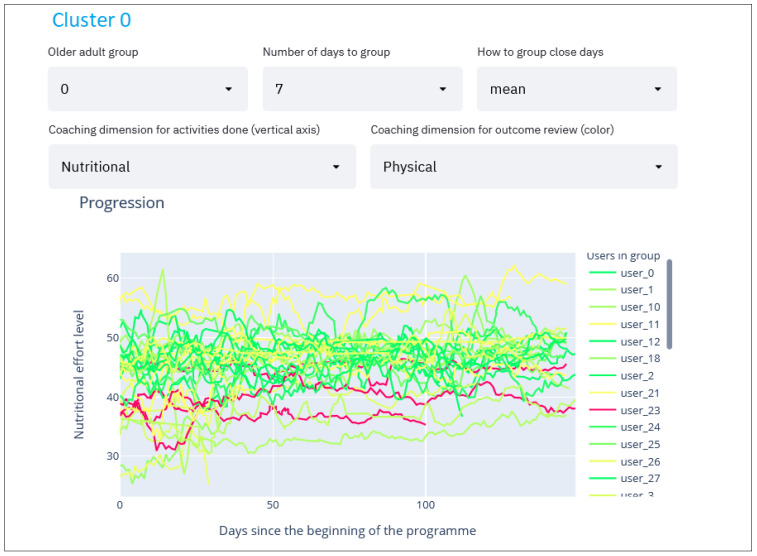
Evolution graph of Group 0. Showing the nutritional activity level and coloring by physical improvement.

**Figure 10 sensors-21-07991-f010:**
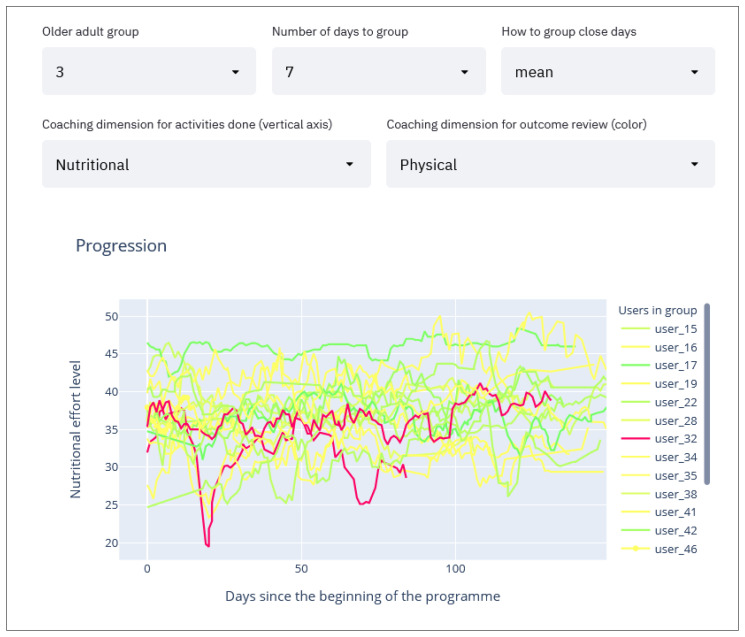
Evolution graph of Group 3. Showing the nutritional activity level and coloring by physical improvement.

**Figure 11 sensors-21-07991-f011:**
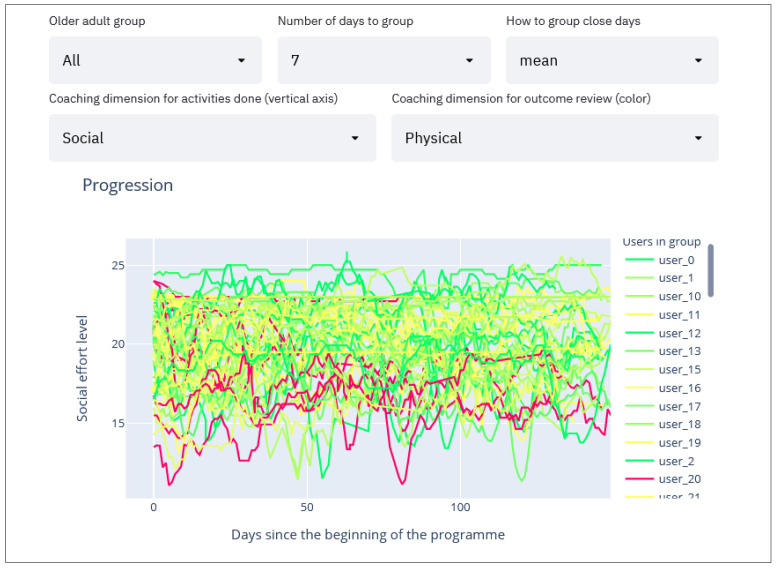
Evolution graph of all participants. Showing the social activity level and coloring by physical improvement.

**Table 1 sensors-21-07991-t001:** This table shows the results of the assessment carried out on the participants at the beginning of the pilot period, giving the mean values and 95% confidence intervals of each observed variable.

Coaching Dimensions	Variable		Mean, (95% CI)
Physical	Chair Stand		12.5, (11.4, 13.6)
	Arm Curl		16.54, (15.33, 17.75)
	Two Minute Step		66.01, (59.93, 72.09)
	Chair Sit and Reach	1	−5.29, (−8.21, −2.37)
		2	−3.58, (−6.55, −0.61)
	Back Scratch	1	−10.33, (−14.01, −6.65)
		2	−9.34, (−12.62, −6.06)
	Foot Up and Go	1	6.55, (6.13, 6.97)
		2	6.11, (5.73, 6.49)
Cognitive	MoCA		25.46, (24.76, 26.16)
	MSC-A		3.22, (2.68, 3.76)
	MAI		6.16, (5.89, 6.43)
Nutritional	MNA mini		4.23, (3.92, 4.54)

The variables “Chair Sit and Reach”, “Back Scratch” and “Foot Up and Go” have been tested two times (1, and 2 in Table 1). In COLAEVA, the best result has been used.

**Table 2 sensors-21-07991-t002:** Daily questionnaire.

Coaching Dimensions	Question		Possible Answers	Weights
Physical	How much light physical activity have you performed? *		None/Less than 1 h/1–2 h/2–3 h/3–4 h/more than 4 h	1
	How much moderate physical activity have you performed? *			2
	How much vigorous physical activity have you performed? *			3
Cognitive	Have you been outside your house?		Yes/No	1
	Have you eaten one or more meals outside of home?			
	Have you been to a shop?			
	Have you prepared your own food at least once?			
	Have you practiced an artistic pastime?			
	Have you read anything?			
Social	How many people have you met?		Number (max 4)	3
	How many people have you talked on the phone with?			2
	How many people have you texted?			1
	Have you been outside your house?		Yes/No	2
	Have you been to a shop?			2
	Consider the social interactions you had, what would you prefer?		Increase a lot Increase a little Not change Decrease a little Decrease a lot	2
Nutritional	Have you eaten one or more meals outside of home?		Yes/No	1
	Have you prepared your own food at least once?			1
	How many times have you added sugar to anything you have eaten?		Number (max 4)	−1
	How many times have you added salt to anything you have eaten or drunk?			−2
	How much water did you have? (unit: water glasses)			2
	How many units of alcohol did you have?			−2
	How many times did you eat something during the day?			1
	How satisfied were you when you stopped eating?		I ate too little/I was just satisfied/I ate slightly too much/I ate too much	3
	Did you have more or less of each of the food groups shown in the picture below? *	Vegetables and Fruits	None/Less/Equal/More/All	2
		Dairy Products		1
		Bread, Cereals and Potatoes		1
		Oils		1
		Meat, Fish and Eggs		2
	What have you eaten?		Complete meal/Light meal/Nothing	3

* These questions were asked 3 times a day, morning/afternoon/evening, or breakfast/lunch/dinner.

**Table 3 sensors-21-07991-t003:** Participant’s exposure to digital technologies. Answers are sorted in decreasing order.

Id	Questions	Sorted Answers									Mean (std)
Q1	Average computer usage per week in hours	60	60	50	50	50	40	35	30	10	42.8 (16)
Q2	Level of computer literacy	5	4	4	4	4	4	3	3	3	3.8 (0.67)
Q3	Familiarity with visual analytics applications	4	4	4	3	3	3	3	2	2	3.1 (0.8)

**Table 4 sensors-21-07991-t004:** SUS questionnaire’s answers sorted in decreasing order.

Id	Questions	Sorted Answers									Mean (std)
Q1	I think that I would like to use this application frequently	5	4	4	4	4	4	4	3	3	3.9 (0.6)
Q2	I found this application unnecessarily complex	4	4	3	2	2	2	2	2	1	2.4 (1)
Q3	I thought this application was easy to use	5	5	4	4	4	4	4	3	3	4.0 (0.7)
Q4	I think that I would need assistance to be able to use this application	4	4	3	3	3	2	2	2	1	2.7 (1)
Q5	I found the various functions in this application were well integrated	5	5	5	4	4	4	4	4	3	4.2 (0.67)
Q6	I thought there was too much inconsistency in this application	3	3	2	2	2	2	1	1	1	1.9 (0.78)
Q7	I would imagine that most people would learn to use this application very quickly	4	4	4	3	3	3	3	3	3	3.3 (0.5)
Q8	I found this application very cumbersome awkward to use	2	2	2	2	2	1	1	1	1	1.6 (0.52)
Q9	I felt very confident using this application	4	4	4	4	4	4	4	3	2	3.7 (0.7)
Q10	I needed to learn a lot of things before I could get going with this application	5	3	2	2	2	2	1	1	1	2.1 (1.27)
Q0	SUS general estimation (mean 71.1)	82.5	77.5	77.5	75	72.5	72.5	72.5	62.5	47.5	71.1 (10.4)

**Table 5 sensors-21-07991-t005:** Personalized questionnaire’s answers sorted in decreasing order.

Id	Questions	Sorted Decreasingly									Mean (std)
Q1	I think that the way users are grouped is easy to understand	5	5	5	5	4	4	4	4	4	4.4 (0.53)
Q2	I find that the graphs really help me differentiate clearly groups of similar users	5	5	5	5	4	4	4	4	2	4.2 (0.97)
Q3	I think this application shows valuable graphs to better understand the user’s evolution	5	5	4	4	4	4	4	4	4	4.2 (0.44)
Q4	I think that I would use this application in my daily basis routine job if I would need to follow up the health status of older adult people	5	5	5	4	4	3	3	3	2	3.8 (1.09)

## Data Availability

The code is available in an open GitHub repository [34]. It makes use of the anonymized data which is also located in this repository. Moreover, we have deployed on Heroku, so that anyone who wants to try COLAEVA [7] can access it.

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
