# Peer review of "COLAEVA: Visual Analytics and Data Mining Web-Based Tool for Virtual Coaching of Older Adult Populations"

_sensors, 2021, doi:10.3390/s21237991_

Round 1
Reviewer 1 Report
The authors present a very interesting issue in the field of improving the quality of life of the elderly. It is especially important due to the progressive aging of the society. The authors in their article present an interesting web application written in Python. This application responds to the challenges posed in the subject of the article and is intended to help in planning and analyzing coaching plans for the elderly.
Although the research conducted by the authors certainly deserves recognition, the indicated number of 65 as the number of people surveyed to obtain the necessary data to create an application, and then only 9 people for testing, seems to be a fairly small reference group, to draw far-reaching conclusions.
Certainly, what has been presented by the authors is characterized by a good research technique and appropriate analysis.
It is interesting and worth emphasizing that the application, apart from the service part, has a module for collecting and analyzing data, which in fact means an unlimited inflow and analysis of data throughout the life of the discussed application. From this point of view, it is an excellent research tool.
The authors' work is easy to read and with interest, although in my opinion the discussion chapter is only an extension of the conclusions. Definitely, it is not a discussion but a collection of achievements, a summary.
Tech: table 4: last column, use not '4' but '4.0'
At the same time, maybe it is worth to align some cells with numbers in tables to the right for easier analysis of numerical data.
In my opinion, the article is worth publishing.
Author Response
Dear Editors,
We want to thank the reviewers for their valuable time and useful contributions. We greatly appreciate the given feedback which will certainly help improve our manuscript.
We have edited the manuscript to address their concerns. Nevertheless, please, do not hesitate to contact us for any further comments or suggestions.
Thank you again for your review and we look forward to hearing from you soon regarding our submission.
Below we respond to the comments you have made. Your comments are in italics, followed by our answers. Please note that the lines that we mention that we have modified something in the article, are with the track changes expanded (the numbering of the lines changes depending on this).
Comments and Suggestions for Authors
The authors present a very interesting issue in the field of improving the quality of life of the elderly. It is especially important due to the progressive aging of the society. The authors in their article present an interesting web application written in Python. This application responds to the challenges posed in the subject of the article and is intended to help in planning and analyzing coaching plans for the elderly.
Thank you
Although the research conducted by the authors certainly deserves recognition, the indicated number of 65 as the number of people surveyed to obtain the necessary data to create an application, and then only 9 people for testing, seems to be a fairly small reference group, to draw far-reaching conclusions.
As you comment below, this tool is designed to be able to load new patient data, so as new patients are added, more robust results will be obtained. We are aware that the current number of patients is not sufficiently high to produce broad recommendations. This point is also discussed in the discussion part.
Although the number of 9 subjects for the usability test may appear low, this number is considered sufficient, taking into account the recommendation by usability experts of five participants to maximize cost/benefit of the evaluations[1]. According to these experts, 80% of the problems of a software application can be identified by only five participants[2]. To avoid this kind of misunderstanding by the reader, we have added to the manuscript a paragraph explaining this (lines 450-453).
Certainly, what has been presented by the authors is characterized by a good research technique and appropriate analysis.
Thank you, we appreciate your positive comment.
It is interesting and worth emphasizing that the application, apart from the service part, has a module for collecting and analyzing data, which in fact means an unlimited inflow and analysis of data throughout the life of the discussed application. From this point of view, it is an excellent research tool.
Thank you for your support, we are very glad to hear that this aspect has been well received..
The authors' work is easy to read and with interest, although in my opinion the discussion chapter is only an extension of the conclusions. Definitely, it is not a discussion but a collection of achievements, a summary.
Following your comment, we agreed that there was a part of summary of the results of usability test. For this reason, we have created an entirely new chapter within the usability test section in which we have inserted these findings (4.4 Usability Test Findings). We have also modified the discussion to be more in line with the section. For example, we have defined how it fits into the project in which it has been elaborated and how it improves it. In addition, the same problem that you have previously commented about the size of the dataset, we have addressed it in this improved section.
Tech: table 4: last column, use not '4' but '4.0'
We have corrected this numbering.
At the same time, maybe it is worth to align some cells with numbers in tables to the right for easier analysis of numerical data.
We agree, we have made this change.
In my opinion, the article is worth publishing.
Thank you, we are thrilled that you value the content and feel that it is worthy of publication.
[1] Nielsen, J.; Landauer, T.K. A mathematical model of the finding of usability problems. In CHI ’93: Proceedings
of the SIGCHI Conference on Human Factors in Computing Systems; ACM Press: New York, NY, USA, 1993;
pp. 206–213.
[2] Krug, S.; Matcho, M. Rocket Surgery Made Easy: The Do-It-Yourself Guide to Finding and Fixing Usability Problems;
New Riders: Indianapolis, IN, USA, 2010.

Reviewer 2 Report
Your paper could improve in a number of areas such as a more thorough discussion of the design, development, testing and evaluation results; clarifying the key significance of the research contribution; ascertaining that the research fits the aims and scope of the journal; and a better command and flow of English writing throughout the paper.
The references should be updated with the most recent in your paper's research field of relevance. I recommend the authors to consult the following survey and empirical papers to contextualize your findings. This should help the readers to understand the novelty of your work.
Machine learning approach to drivers of bank lending: evidence from an emerging economy. Financ Innov 7, 20 (2021). https://doi.org/10.1186/s40854-021-00237-1
Evaluation of Clustering Algorithms for Financial Risk Analysis using MCDM Methods , DOI: HTTP://DX.DOI.ORG/10.1016/j.ins.2014.02.137, Information Sciences, 27(2014):1-12
DAViS: a unified solution for data collection, analyzation, and visualization in real-time stock market prediction. Financ Innov 7, 56 (2021). https://doi.org/10.1186/s40854-021-00269-7
An Integrated Cluster Detection, Optimization and Interpretation Approach for Financial Data, IEEE Transactions on Cybernetics, (2021), DOI: https://doi.org/10.1109/TCYB.2021.3109066
Author Response
Dear Editors,
We want to thank the reviewers for their valuable time and useful contributions. We greatly appreciate the given feedback which will certainly help improve our manuscript.
We have edited the manuscript to address their concerns. Nevertheless, please, do not hesitate to contact us for any further comments or suggestions.
Thank you again for your review and we look forward to hearing from you soon regarding our submission.
Below we respond to the comments you have made. Your comments are in italics, followed by our answers. Please note that the lines that we mention that we have modified something in the article, are with the track changes expanded (the numbering of the lines changes depending on this).
Comments and Suggestions for Authors
Your paper could improve in a number of areas such as a more thorough discussion of the design, development, testing and evaluation results; clarifying the key significance of the research contribution; ascertaining that the research fits the aims and scope of the journal; and a better command and flow of English writing throughout the paper.
The references should be updated with the most recent in your paper's research field of relevance. I recommend the authors to consult the following survey and empirical papers to contextualize your findings. This should help the readers to understand the novelty of your work.
Machine learning approach to drivers of bank lending: evidence from an emerging economy. Financ Innov 7, 20 (2021). https://doi.org/10.1186/s40854-021-00237-1
Evaluation of Clustering Algorithms for Financial Risk Analysis using MCDM Methods , DOI: HTTP://DX.DOI.ORG/10.1016/j.ins.2014.02.137, Information Sciences, 27(2014):1-12
DAViS: a unified solution for data collection, analyzation, and visualization in real-time stock market prediction. Financ Innov 7, 56 (2021). https://doi.org/10.1186/s40854-021-00269-7
An Integrated Cluster Detection, Optimization and Interpretation Approach for Financial Data, IEEE Transactions on Cybernetics, (2021), DOI: https://doi.org/10.1109/TCYB.2021.3109066
Authors answer
The line number is different depending on whether you have extended track changes or not. The numbers that we indicate position in the answers are with the track changes extended.
The main contributions or objectives of COLAEVA are listed in the introduction in lines 63-70. In addition, in the following paragraph, we explain its characteristics and the steps we have followed. Additionally, we have changed the structure of the discussion section, in case it could led to confusion. We have emphasized more which points the study in this paper helps (lines 559-580), and we have moved some findings that we had defined to a separate section in the usability test (lines 533-558). We hope this will make the content more accessible to the reader.
We have explicitly stated how our paper fits into the special issue topics and context in the Introduction section (lines 71-73)
After receiving feedback regarding language errors, two of the co-authors who are native English speakers (American and British) performed another exhaustive review of the paper. If any further grammatical errors remain, please indicate them and we will be sure to make those changes.
Based on the examples you have provided us, we have found some articles which are aligned with the paper topic. We have added these papers in the introduction. This content has been added in lines 85-90.

Round 2
Reviewer 2 Report
Your paper could improve in a number of areas such as a more thorough discussion of the design, development, testing and evaluation results; clarifying the key significance of the research contribution; ascertaining that the research fits the aims and scope of the journal; and a better command and flow of English writing throughout the paper.
The references should be updated with the most recent in your paper's research field of relevance. I recommend the authors to consult the following survey and empirical papers to contextualize your findings. This should help the readers to understand the novelty of your work.
An Integrated Cluster Detection, Optimization and Interpretation Approach for Financial Data, IEEE Transactions on Cybernetics, (2021), DOI: https://doi.org/10.1109/TCYB.2021.3109066
DAViS: a unified solution for data collection, analyzation, and visualization in real-time stock market prediction. Financ Innov 7, 56 (2021). https://doi.org/10.1186/s40854-021-00269-7
Comprehensive review of text-mining applications in finance. Financ Innov 6, 39 (2020). https://doi.org/10.1186/s40854-020-00205-1